# Lipidomic Profiling Reveals Distinct Differences in Sphingolipids Metabolic Pathway between Healthy *Apis cerana cerana larvae* and Chinese Sacbrood Disease

**DOI:** 10.3390/insects12080703

**Published:** 2021-08-05

**Authors:** Xiaoqun Dang, Yan Li, Xiaoqing Li, Chengcheng Wang, Zhengang Ma, Linling Wang, Xiaodong Fan, Zhi Li, Dunyuan Huang, Jinshan Xu, Zeyang Zhou

**Affiliations:** 1Chongqing Key Laboratory of Vector Insect, College of Life Science, Chongqing Normal University, Chongqing 401331, China; 20132103@cqnu.edu.cn (X.D.); 2018110513030@stu.cqnu.edu.cn (Y.L.); 2019110513029@stu.cqnu.edu.cn (X.L.); 2020110513043@stu.cqnu.edu.cn (C.W.); 20140003@cqnu.edu.cn (Z.M.); 20130776@cqnu.edu.cn (L.W.); fanxd@cqnu.edu.cn (X.F.); lizhi@cqnu.edu.cn (Z.L.); 20170054@cqnu.edu.cn (D.H.); 2State Key Laboratory of Silkworm Genome Biology, College of Biotechnology, Southwest University, Chongqing 400715, China; 3Chongqing Key Laboratory of Microsporidia Infection and Control, Southwest University, Chongqing 400715, China

**Keywords:** Chinese sacbrood virus, *A. cerana cerana* larva, lipidomics, glycerolipids, sphingolipid

## Abstract

**Simple Summary:**

Chinese Sacbrood Virus (CSBV) is one of the most destructive viruses; it causes Chinese sacbrood disease (CSD) in *Apis cerana cerana*, resulting in heavy economic loss. In this study, the first comprehensive analysis of the lipidome of *A. c*. *cerana* larvae-infected CSBV was performed. Viruses rely on host metabolites to infect, replicate and disseminate from several tissues. The metabolic environments play crucial roles in these processes amplification and reproduction processes. Using high-resolution mass spectrometry, we profiled 2164 lipids in larva samples obtained from healthy controls and larvae with CSBV infection. A total of 266 lipids, including categories of DG, TG, PC, PE, SM and Cer, were significantly changed after CSBV infection, considering that qRT-PCR showed that seven sphingolipid enzymes involved in the sphingolipid pathway were increased. This suggests that the lipidome of honeybee larvae is altered in CSBV infection, and sphingolipids may contribute to CSD progression. Specific changes in the lipidome such as TG (39:6), which showed a more than 10-fold increase, may be possible targets for the detection of CSD. Lipid pathways analyses revealed that glycerophospholipid metabolism was the most enriched pathway and may be associated with CSBV infection.

**Abstract:**

Chinese sacbrood disease (CSD), which is caused by Chinese sacbrood virus (CSBV), is a major viral disease in *Apis cerana cerana* larvae. Analysis of lipid composition is critical to the study of CSBV replication. The host lipidome profiling during CSBV infection has not been conducted. This paper identified the lipidome of the CSBV–larvae interaction through high-resolution mass spectrometry. A total of 2164 lipids were detected and divided into 20 categories. Comparison of lipidome between healthy and CSBV infected-larvae showed that 266 lipid species were altered by CSBV infection. Furthermore, qRT-PCR showed that various sphingolipid enzymes and the contents of sphingolipids in the larvae were increased, indicating that sphingolipids may be important for CSBV infection. Importantly, Cer (d14:1 + hO/21:0 + O), DG (41:0e), PE (18:0e/18:3), SM (d20:0/19:1), SM (d37:1), TG (16:0/18:1/18:3), TG (18:1/20:4/21:0) and TG (43:7) were significantly altered in both CSBV_24 h vs. CK_24 h and CSBV_48 h vs. CK_48 h. Moreover, TG (39:6), which was increased by more than 10-fold, could be used as a biomarker for the early detection of CSD. This study provides evidence that global lipidome homeostasis in *A. c. cerana* larvae is remodeled after CSBV infection. Detailed studies in the future may improve the understanding of the relationship between the sphingolipid pathway and CSBV replication.

## 1. Introduction

*Apis mellifera* and *Apis cerana cerana*, which are required for agricultural production and nutrient-rich beehive products, are increasingly grown in China [1]. However, over the last few decades, honeybee populations have been found to often die or escape because of infection with pathogens such as viruses, bacteria, fungi, parasites and protozoa [2,3]. Sacbrood virus (SBV) is distributed worldwide and is the first honeybee virus identified by Bailey [4,5]. Infection by SBV is lethal to honeybee larvae, which results in entire colony collapse for *A. c*. *cerana,* while *A. mellifera*-infected colonies rarely collapse [6,7]. Thus, the Chinese sacbrood virus (CSBV) is the most serious threat to *A. c. cerana* health. Currently, CSBV is widespread in China and Southeast Asia worldwide [8].

CSBV mainly infects 1- to 3-day-old larvae of *A. c**. cerana* by breeding and results in larval death and reduces the lifespan of adult bees [6]. Symptoms of CSBV infection include ecdysial fluid accumulation, cuticle discoloration and formation of “larvae sac” before death [9]. CSBV has a positive-sense single-stranded RNA genome and is a typical member of the genus Iflavirus, the family Iflaviridae and the order Picornavirales [10]. The genome encodes a single 2858-aa-long polyprotein that is cotranslationally and posttranslationally cleaved by viral proteases into functional protein subunits. To date, three iflaviruses have been structurally characterized: slow bee paralysis virus (SBPV), deformed wing virus (DWV) and SBV [11,12,13]. The structural analysis of SBV showed that the pore of an empty virion particle expands at pH 5.8 compared with the pore of full virion at pH 7.4 (12 Å vs. 7 Å in diameter), and subsequently, the genome is released [14].

RNA viruses need host factors to support their replication, and host lipid components are necessary. It is reported that (+) RNA viruses remodel lipid metabolism through coordinated virus–host interactions to create a suitable microenvironment to survive and thrive in host cells [15]. For example, the Hepatitis C virus (HCV) life cycle is tightly linked to the host cell lipid metabolism [16,17]. HCV-infected cells accumulate cholesterol, phospholipids and polyunsaturated fatty acids. Host lipidome analysis during rhinovirus replication showed increased fatty acid elongation and desaturation [18]. Serum lipidome analysis after Zika virus (ZIKV) infection demonstrated an increase in several phosphatidylethanolamine (PE) lipid species, suggesting a link between ZIKV life cycle and peroxisomes [19]. Dengue virus (DENV) is an arbovirus transmitted by mosquitoes, such as *Aedes aegypti* and *Aedes albopictus*. The midgut metabolomics of *A*. *aegypti* infected with DENV revealed that glycerophospholipids, sphingolipids and fatty acyls were coincident with the kinetics of viral replication [20,21]. Collectively, these data demonstrate that lipids play critical roles in viral infection.

Although lipid biochemistry has been studied in insects for several years, whether the host lipids regulate CSBV replication remains unknown. Thus, identification and analysis of lipid components of the host are important for the investigation of the molecular mechanism of CSBV replication. In this study, we used high-resolution mass spectrometry to explore metabolic changes in the larvae of *A. c*. *cerana* exposed to CSBV-containing foods. Using a time-course study, we compared lipid profiles of infected and uninfected larvae at 24 h and 48 h post-infection (peak viral replication). Specific lipids, including diacylglycerol, triglycerides and ceramide, after virus infection were identified, which represented the lipid species that were regulated by CSBV infection. Furthermore, qRT-PCR analysis revealed that sphingolipid was the most perturbed pathway after CSBV infection.

## 2. Materials and Methods

### 2.1. Sample Collection

For virus purification, the 5th-instar larvae with CSBV infection were collected and placed in sterile PBS. Following the methodology described by Feng et al. [22], CSBV was purified and filtered through a 0.22 μm cell filter and stored at −80 °C. Subsequently, we collected larvae from colonies and transferred them to 24-well plate. The larvae were then reared in an incubator at 34 ± 1 °C and 90% ± 5% relative humidity as previously described [23]. After 12 h incubation, the purified CSBV was fed to 2-day-old *A. c*. *cerana* larvae with 2 × 10^5^ copies. For each treatment-sampling time point, one group of 5–6 larvae were collected and washed with DEPC water and put into a 0.5 mL microfuge tube, then immediately flash-frozen in liquid nitrogen and stored at −80 °C for lipidomics analysis. Each experiment was repeated 3 times. The rest samples were used for qRT-PCR analysis.

### 2.2. Lipidomics

For lipids extract, 200 μL ddH_2_O was added to 100 mg larvae for homogenate, then 240 μL pre-cooled methanol was added and dissolved, after which 800 μL MTBE was added for mixing. The mixture was subjected to ultrasonic in 4 °C water bath for 20 min and incubated at room temperature for 30 min. Last, the upper organic phase was obtained by 14,000× *g* centrifugation 15 min at 10 °C and dried with the protection of nitrogen blowing. Before mass spectrometry analysis, the solid residue was re-dissolved with 200 μL 90% isopropyl alcohol/acetylene solution and then centrifuged at 14,000× *g* for 15 min at 10 °C. The supernatants were used for UPLC-Q-Orbitrap MS analysis. Before lipidomics analysis, quality control samples (QC) were prepared by taking equal amounts of samples from each group and mixing them for QC. QC samples were used to determine the state of the instrument before injection and balance the chromatography-mass spectrometry system, and then they were used to evaluate the stability of the system throughout the experiment.

Samples were isolated using the UHPLC Nexera LC-30A ultra-efficient liquid chromatography system. The column was maintained at a temperature of 45 °C and eluted at a flow rate of 0.3 mL/min. The injection volume was 3 uL. The mobile phase was composed of A (6:4 acetonitrile:water, with 10 mM ammonium acetate) and B (1:9 acetonitrile:isopropanol, with 10 mM ammonium acetate) with a linear gradient elution: 0–2 min, 30% B; 2–25 min, 30–100% B; 25–30 min, 30% B. Lipid extracts were subjected to mass spectrometric analysis using a Q Exactive mass spectrometer (Thermo Fisher Scientific, Waltham, MA, USA) equipped with a TriVersa NanoMate (Advion Biosciences, Ithaca, NY, USA). The source parameters were set as follows: capillary voltage 3.00 KV for ESI+ and 2.50 KV for ESI-, (other parameters of both positive and negative ions were the same) source temperature 300 °C. The acquisition cycle consisted of FT MS and FT MS/MS scans in positive and negative ion mode. FT MS spectra were acquired at the mass resolution of Rm/z 200 = 70,000, and FT MS/MS spectra were acquired at the mass resolution of Rm/z 200 = 17,500.

### 2.3. Data Processing, Lipid Identification and Quantification

Lipid species detected by high-resolution FT MS and FT MS/MS analysis with LipidSearch software version 4.1 (Thermo™) (Thermo Science, Waltham, MA, USA) and quantified using Excel-based calculation. In short, the main parameters were as follows: precursor tolerance, 5 ppm; product tolerance, 5 ppm; product ion threshold, 5%. Then, the data matrix was imported to the SIMCA-P 14.1 software package (Umetrics, Umea, Sweden) to conduct unsupervised component analysis (principal component analysis PCA), partial least-squares-discriminant analysis (PLS-DA) and orthogonal partial least-squares-discriminant analysis (OPLS-DA). Combined student T test, Fold Change Analysis (FC) and the variable importance in projection (VIP), which reflects both the loading weights for each component and the variability of the response explained by this component, were used to find the potential markers in the infected group. (|Log2 fold change|) >1 and *p*-value < 0.05 were used to screen the different lipids features among different groups.

### 2.4. Statistical Analysis

Data were expressed at the form of means ± SD. Univariate statistical analysis was performed using SPSS (version 19). Data were analyzed using an independent *t*-test. *p*-values < 0.05 were considered statistically significant.

### 2.5. Quantitative Real-Time PCR

To investigate CSBV proliferation in *A. cerana cerana* larvae, the specific primer pair of the CSBV VP1 gene was designed. The primers for sphingolipid pathway-related enzymes used in quantitative RT-PCR (qPCR) were also designed based on their sequences, downloaded from NCBI (Table 1). Total RNAs were extracted with Trizol, and cDNA was acquired by EvoScript Universal cDNA (Cat No. 07912374001, Roche, Mannheim, Germany). The qPCR reaction was performed in Bio-Rad CFX96 real-time system (Bio-Rad, Hercules, CA, USA) using FastStart essential DNA Green Master (Cat No. 06402712001, Roche) according to the instructions from the manufacturer. Briefly, 10 ng of cDNA, 10 μL Master mix 2×conc, 0.5 μL of each primer and ddH_2_O were added to make a 20 μL reaction, and PCR was performed under the following conditions: denaturation at 95 °C for 30 s, 30 cycles of 95 °C for 5 s, 55 °C for 30 s and 72 °C for 50 s. The actin gene of *A. c*. *cerana* was used as an internal gene to normalize the target gene expression and to correct the variation of sample-to-sample. The relative expression levels of sphingolipid enzymes and CSBV VP1 were calculated based on the relative quantitative method (2-ΔΔt). To detect the copy number of CSBV, VP1 was cloned into vector pMD19T, and the constructed standard plasmid was detected for Ct value of copy number by real-time fluorescence quantitative PCR and then a standard curve of virus copy number against Ct was plotted, and the corresponding equation was obtained Statistical analyses were performed using the GraphPad Prism8 software (https://www.graphpad.com/scientific-software/prism/, accessed on 11 November 2019). A two-way analysis of variance with a least significant difference test was applied to compare differences in gene expression among multisamples. The level of significant difference was set at *p* < 0.05.

## 3. Results and Discussion

### 3.1. General Description of Lipidome Data

The morphology of CSBV-infected larvae displayed a change in body color from white to light yellow. In addition, the infected body was swollen with ecdysial fluid accumulation under the transparent epidermis (Figure 1A,B). Subsequently, individual groups were tested for CSBV VP1 by qRT-PCR (Figure 1C) after samples were collected to estimate virus infection per group. The relative transcript level of VP1 was undetectable in healthy larvae. In contrast, a significant increase in the virions was found at 12 h, 24 h and 48 h post-infection. Both the symptoms of CSD and the copies of CSBV indicated that the lipidome profile of the larvae could represent the infected state.

The lipids extracted from each time point infected larvae were subjected to both negative and positive electrospray ionization modes to obtain the most comprehensive coverage. After pareto scaling, the peaks extracted from all the experimental samples and QC samples were analyzed by PCA. As shown in Figure 2A, QC samples were clustered in the middle of each group, which indicated that the experiment had good repeatability. The PCA model parameters were obtained by cross-validation (seven cycles of interactive verification). The obtained lipid profile differences reflect the biological differences between CSBV infection and healthy larvae. Partial least squares discrimination analysis (PLSDA) was used to visually discriminate between CSBV-infected larvae and healthy controls. As shown in Figure 2B, CSBV-infected groups (CSBV_24 h and CSBV_48 h) could be reliably discriminated from healthy controls.

For all the samples, a total of 2164 molecular features from all modes and time points post-infection were detected (Figure 3A). These features were subjected to statistical analysis to compare the differences in metabolite abundance between CSBV-infected and uninfected at each time point post-infection. A total of 266 features that showed statistically significant differences in abundance were identified using LIPID MAPS databases. These 266 lipid molecular species were distributed into ceramide (Cer), lysophosphatidylcholine (LPC), lysophosphatidylethanolamine (LPE), lysophosphatidylglycerol (LPG), monoglyceride (MG), phosphatidate (PA), phosphatidylcholine (PC), phosphatidylethanolamine (PE), diglyceride (DG), phosphatidylglycerol (PG), phosphatidylinositol (PI), phosphatidylserine (PS), sphingomyelin (SM), sphingolipids (So), triglyceride (TG) and (O-acyl)-1-hydroxy fatty acid (OAHFA) (Figure 3A). Among the features with altered levels, 75 and 191 features showed differential abundance at 24 h and 48 h post-infection, respectively, whereas 10 features showed differential abundance at both 24 h and 48 h post-infection (Figure 3B).

The majority of the metabolites were unchanged (with <1-fold intensity changes upon infection, *p* > 0.05). In addition, about 12% of features (266 out of 2164) with differential abundance were identified using three metabolite databases, and the putative identifications were categorized into different metabolite classes. The overall trend of lipid molecular levels that changed upon CSBV infection is summarized in Figure 3C, while the majority of those (182 out of 266 features) that showed greater than 1-fold changes in intensity had higher abundances upon CSBV infection (Figure 3B). Only 28.6% of lipid species decreased in abundance during CSBV infection.

To observe the discrimination trend in more detail, a hierarchical clustering analysis was performed based on the degree of similarity of lipid abundance profiles to show the overall trend of all significant ion features. As indicated in Figure 4, most of the significant features expressed an up-regulation trend after both 24 h (Figure 4A) and 48 h post-infection (Figure 4B) compared with the healthy larvae. Moreover, the larvae of *A. c**. cerana* are so small that there were inevitable differences in the three repeats because of the individual differences. Further study will focus on midguts, which is the main infection site.

### 3.2. Glycerolipids Showed Maximal Remodeling after CSBV Infection

To gain insight into the lipid metabolic reprogramming of the larvae during CSBV-infection, we first analyzed the glycerolipids (GP) profile during viral replication in comparison to uninfected larvae, which were majorly altered (about 180 of 266). GPs, such as mono-, di- and triacylglycerols (MG, DG and TG), are critical effectors of energy metabolism in insects and mammals. DG is a critical second messenger regulating cell proliferation, survival, mitochondrial physiology, gene expression and apoptosis [24,25]. TG containing a glycerol backbone and three fatty acyl chains has a high energy content. Most of the DG and TG levels of CSBV-infected larvae were higher than those of healthy larvae (Figure 5C). A total of 122 species of TGs were identified in this study, of which 113 TGs were significantly changed in abundance upon infection (Figure 5). Moreover, 72 species of TG levels were higher during the early stage (24 hpi) in the CSBV-infected larvae, indicating that TGs may serve as storage molecules, induced by the virus as an energy and lipid pool to support the demand for viral replication [26,27]. HCV is known to promote the mobilization and recruitment of lipid droplets responsible for the cellular stock of TGs [28,29]. In addition, phospholipase B1 (PLB1) acts as a hydrolyzed enzyme, which can remove fatty acids from phospholipids and hemolytic phospholipids [30]. qRT-PCR showed that AcPLB1 was significantly increased by nine times at 24 h post-CSBV infection (Figure 5D), presumably causing glycerophospholipid degradation, which may contribute to the accumulation of DG and TG. GPs are critical effectors of energy metabolism in insects and mammals. Given that the transcriptional level of AcPLB1 was up-regulated, CSBV infection may cause a higher level of lipolysis. The precise role of the increased TG accumulation needs to be explored.

### 3.3. Alterations in Phospholipids Homeostasis during CSBV Infection

Besides the GPs, phospholipids including PA, PC, PE, PS, PI, PG, LPC, LPG and OAHFA were identified in higher abundance in the CSBV versus healthy larvae. Selected PC species were up-regulated at both time points (Figure 6). Interestingly, the majority of PC species that were up-regulated had unsaturated fatty acyl chains. Of the PC lipids identified in this study, almost all were increased in CSBV-infected larvae, including 15 with a *p* < 0.05. The only statistically significant PC lipid that was decreased was PC (14:0/22:2) (*p*-value = 0.01) (Figure 6). A significant increase in PC is associated with viral replication [31]. Thus, PC (18:1/14:1) was identified as a characteristic molecule for the group of the infected larvae (Table 2). Analysis of the overall fold-change in PE (Figure 6) showed an increase between the CSBV-infected and control. PE (18:0/18:2) was up-regulated more than 4-fold. Combining PA (38:2) was also up-regulated more than 2-fold; these results indicated that CSBV infection has changed the lipid composition of the host membrane.

PC synthesis is significantly up-regulated by many viral infections, including brome mosaic virus (BMV) [31], Flock House virus (FHV) [32], DENV [20] and poliovirus [33]. Consistent with previous reports, increased levels of PC were identified in CSBV-infected larvae. In addition, PE can be synthesized either by reactions analogous to those of de novo synthesis of PC or by decarboxylation of PS. Furthermore, PE was redistributed to the TBSV replication sites by protein p33 [34]. The increased PEs identified in CSBV-infected larva may contribute to the replication of CSBV.

### 3.4. CSBV Infection Causes an Accumulation of Sphingolipids

Sphingolipids have many biological functions that regulate cell death, growth, differentiation and intracellular trafficking and are critical in microbial pathogenesis [35,36]. De novo synthesis of SPs occurs in the endoplasmic reticulum through the condensation of L-serine and palmitoyl-CoA to form ceramide (Cer) via several intermediates (Figure 7A). The primary sphingolipids regulated during CSBV infection were sphingomyelin (SM) and ceramide (Cer). More recently, sphingolipids and their metabolizing enzymes were implicated in regulating the interactions between bacterial or viral pathogens and plants or animals [37]. A study has reported that HCV increases sphingolipids levels in infected host cells [38]. It has been further demonstrated that the increased Cer and SM are required for WNV replication [39]. Inhibition of Serine palmitoyltransferase (SPT), the first-step enzyme in Cer biosynthesis, suppressed HCV replication [40,41]. Several viruses such as measles virus, rhinovirus (RV) and DENV [42] activated sphingomyelinase (SMase), which is responsible for the degradation of SM and synthesis of Cer. Sphingomyelin synthase (SMS) is also required for HIV-mediated membrane fusion by co-localizing with the HIV receptor in the plasma membrane [43]. Our study identified 492 species of sphingolipids in *A.*
*c*. *cerana* with 62 species that were altered upon CSBV-infection. Cer (d20:1), Cer (d17:0/16:0) and SM (d41:2) were up-regulated in CSBV-infected larvae by >2-fold compared to the control (Figure 8, Table 2). Similarly, SM was also up-regulated in CSBV-infected larvae.

To investigate the relative expression levels of the nine genes encoding sphingolipid-metabolizing enzymes in different CSBV infection stages, qPCR assays were performed using specific primers (Table 1). When comparing CSBV-infected *A.*
*c**. cerana* larvae with CSBV-free larvae, six genes (AcCS5, AcCeramidase, AcSMPD, AcSMPD1, AcSMPD4 and AcGSC) were more highly expressed at 24 h post-infection (Figure 7B), suggesting that the increase in Cer during the early infection is possibly due to both de novo synthesis and SM degradation. After 48 h CSBV infection, AcCS5 was continuously increased (Figure 7B), and AcSMase showed a 6-fold higher transcription level than control (6-fold, *p* = 0.02). In addition, AcSPT1 was more highly expressed (*p* = 0.002), while AcGSC4 decreased. RNA-dependent RNA polymerase (RdRp) of CSBV contains a helix–turn–helix sphingolipid-binding motif in its finger domain. All these data indicate that CSBV may use Cer and SM for replication. However, the exact mechanism of CSBV replication remains unknown and requires further studies.

### 3.5. Biomarkers Screening of the CSBV-Infected Larvae

Based on the criteria of *p*-value < 0.05 and |Log2 fold change| > 1 and VIP >1.5 for the identification of potential biomarkers, OPLSDA analysis detected several potential biomarkers of CSBV infection, as indicated in Table 2. A total of 20 significantly changed lipids were identified when the two groups of samples were compared in pairs. These lipids included those belonging to the phospholipid, glycerolipid and sphingolipid categories. Notably, Cer (d14:1 + hO/21:0 + O), DG (41:0e), PE (18:0e/18:3), SM (d20:0/19:1), SM (d37:1), TG (16:0/18:1/18:3), TG (18:1/20:4/21:0) and TG (43:7) were significantly changed in both CSBV_24 h vs. CK_24 h and CSBV_48 h vs. CK_48 h. Furthermore, TG (39:6) was increased by >10 fold and can be used as a biomarker for early detection of CSD. Fatty acids can be incorporated into complex lipids such as lycerophospholipids and glycerolipids that have structural roles in membranes. In this study, 15 OAHFA were detected, of which three had lower abundances in larvae at 48 h post-infection compared to uninfected controls (Appendix A). OAHFA is a unique type of ULCFA that constitutes the fatty acyl moiety in Cer, as reported in mice with fatty acid transport protein 4 (FATP4) mutations [44].

### 3.6. Pathway Analysis of CSBV-Infected A. c. cerana larvae

Based on the list of significantly up-regulated lipids after CSBV-infection, MetaboAnalyst (http://www.metaboanalyst.ca, accessed on 11 November 2019) was applied to investigate the pathways that were markedly perturbed. As shown in Figure 9, upon CSBV infection, pathways related to glycerophospholipid metabolism, glycosylphosphatidylinositol (GPI) -anchor biosynthesis, glycerolipid metabolism, arachidonic acid metabolism and alpha-linolenic acid metabolism were found to be significantly affected. Pathway impact results indicate that the glycerophospholipid metabolism pathways presented a higher impact than the other pathways. Therefore, we speculated that the disorder of phospholipids metabolism may be associated with CSBV entry.

## 4. Conclusions

CSD is a viral disease of *A.*
*c*. *cerana* larvae. One of its typical symptoms is the formation of a “sac” with ecdysial fluid accumulation. Thus, metabolic disorders of the host caused by CSBV infection provide insight into the phenomenon of the ‘sac’. The lipidome analysis of honeybee larva after CSBV infection in this study demonstrates the complex response patterns in the whole animal. A total of 266 lipids, including categories of DG, TG, PC, PE, SM and Cer, were significantly altered upon CSBV infection. The level of TGs was found to be significantly increased in CSBV infection. Furthermore, TG (39:6) of CSBV_24 h increased by more than 10 fold compared to CK_24 h and may be used as a biomarker for early detection of CSD. Given that AcPLB1 was up-regulated, CSBV infection causes a higher level of lipolysis. The precise role of the increased TG accumulation needs to be explored. Importantly, marked elevation of sphingolipids levels was coincident with the expression levels of eight sphingolipid enzymes upon CSBV infection. Based on these findings, the next steps will be to investigate the mechanisms by which these lipid species play a role in CSBV replication, as well as their potential use as therapeutic targets for CSBV infection.

## Figures and Tables

**Figure 1 insects-12-00703-f001:**
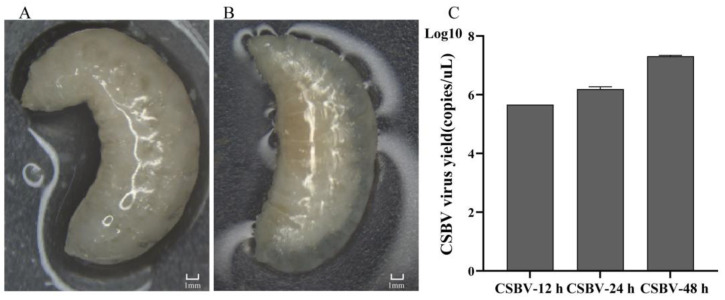
Morphological characteristics of CSBV-infected larvae and relative quantification analysis of CSBV at each time point after larvae infection. The CSBV-infected larva (**B**) was swollen and smaller compared with the healthy larvae of *A. c. cerana* (**A**). (**C**) The copies of the CSBV virus.

**Figure 2 insects-12-00703-f002:**
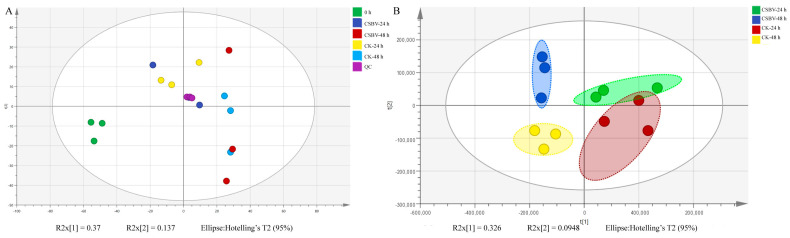
Score plots obtained from PCA (**A**) and PLSDA (**B**) analysis of the present lipidomics data in both CSBV-infected and healthy larvae.

**Figure 3 insects-12-00703-f003:**
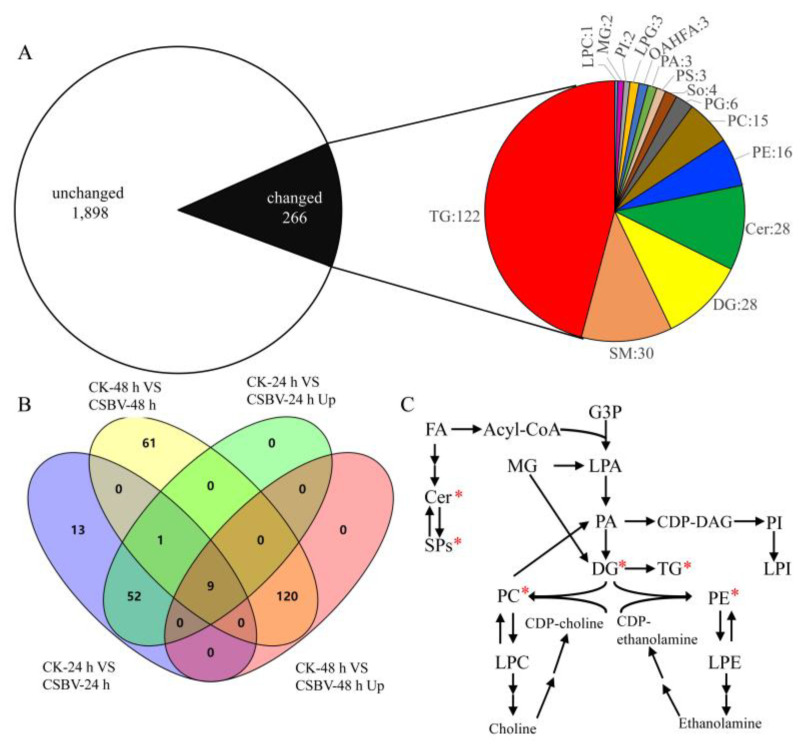
Lipidomics profile of the *A. c**. cerana* larvae during the course of CSBV infection significant changes were observed in the metabolic profile of the larvae upon infection. (**A**) The right pie chart displays the total lipids identified in this study, the left pie chart shows numbers of features in the larvae detected on 24 h and 48 h following a CSBV-infectious compared to the health larvae with significantly altered levels of abundance (|Log2 fold change| > 1 and *p*-value < 0.05) in black and non-significantly altered levels of abundance (|Log2 fold change| < 1 or *p*-value > 0.05) in white. The right pie chart displays the distribution of the changed lipids. (**B**) The Venn diagram shows numbers of features that were altered in abundance in CSBV-infected larvae compared to uninfected larvae. (**C**) Overview of lipid classes observed in this study and their relationships to each other within metabolic pathways. Red stars ‘*’ indicate that the lipids were significantly changed in lipid levels.

**Figure 4 insects-12-00703-f004:**
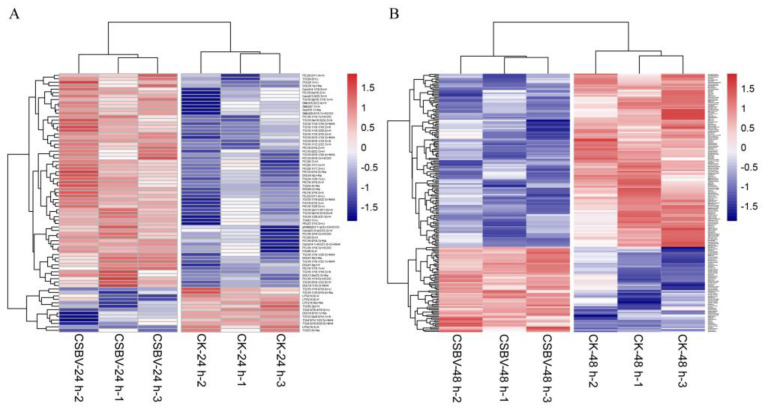
Global lipidome profile of CSBV-infected versus uninfected *A.*
*c**. cerana* larvae. (**A**) Heatmap showing the lipidomic analysis of CSBV_24 h vs. CK_24 h; (**B**) heatmap showing the lipidome analysis of CSBV_48 h vs. CK_48 h.

**Figure 5 insects-12-00703-f005:**
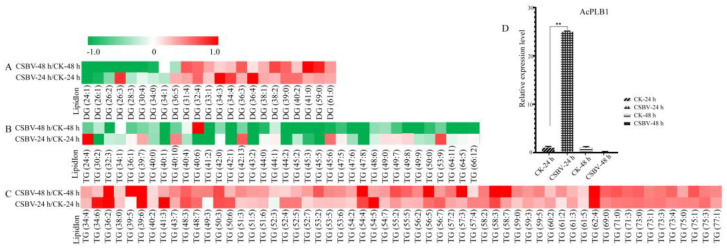
Lipid species of DG and TG were more induced during the CSBV infection process. (**A**) Heatmap showing the DG change of both CSBV_24 h vs. CK_24 h and CSBV_48 h vs. CK_48 h; (**B**) heatmap showing the down-regulation of TG in CSBV_24 h vs. CK_24 h or CSBV_48 h vs. CK_48 h; (**C**) hatmap showing the up-regulation of TG in both CSBV_24 h vs. CK_24 h and CSBV_48 h vs. CK_48 h; (**D**) the expression levels of phospholipidase B1 post infection of CSBV and control by quantitative real-time PCR. ** indicates *p*-value of <0.01.

**Figure 6 insects-12-00703-f006:**
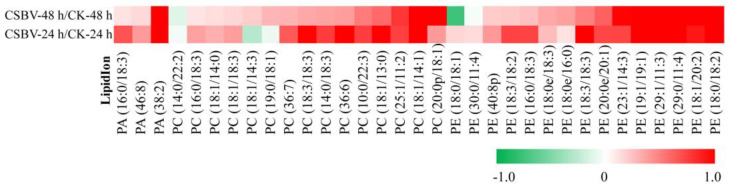
Heatmap of Log2 fold changes in phospholipids species following CSBV infection of *A.*
*c**. cerana larvae* on *p*.i. 24 h and *p*.i. 48 h arranged by putative ID subclasses.

**Figure 7 insects-12-00703-f007:**
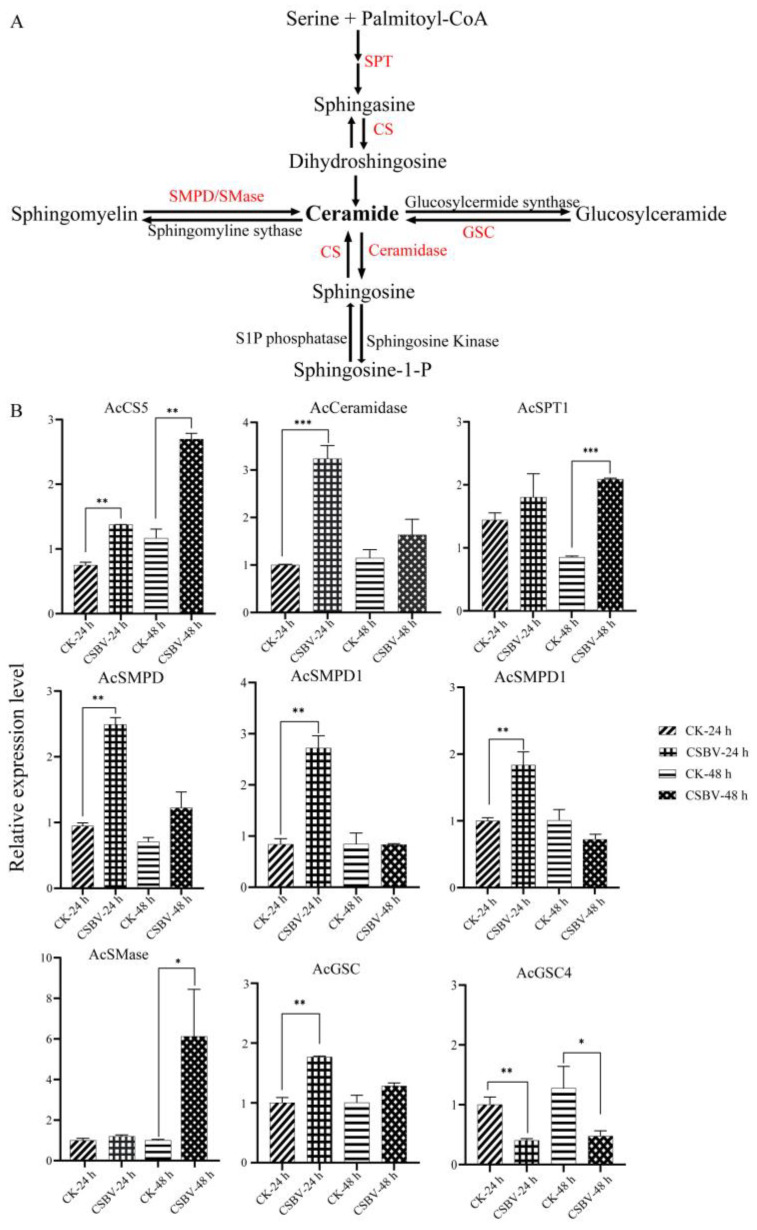
The pathway of sphingolipid metabolism (**A**) and the expression levels of 9 sphingolipid enzymes upon CSBV infection by quantitative real-time PCR (**B**). The enzymes analyzed in this study are in red and full names are listed in Table 1. Data are shown as means ± SE. Asterisks (*) indicates statistically significant *p*-value of <0.05, ** *p*-value of <0.01, *** *p*-value of <0.001.

**Figure 8 insects-12-00703-f008:**
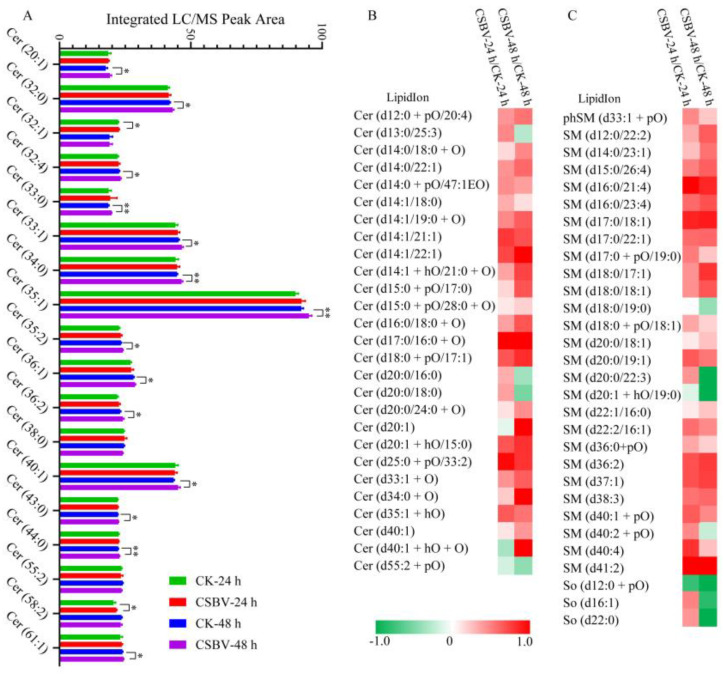
Ceramide and sphingomyelin were regulated under CSBV infection. (**A**) The total relative content of ceramide (Cer) detected in CSBV-infection groups and healthy larvae. (**B**) and (**C**) are heatmaps of Log2 fold changes of ceramide (Cer) and sphingomyelin (SM) species, respectively. Asterisks (*) indicates statistically significant *p*-value of <0.05, ** *p*-value of <0.01.

**Figure 9 insects-12-00703-f009:**
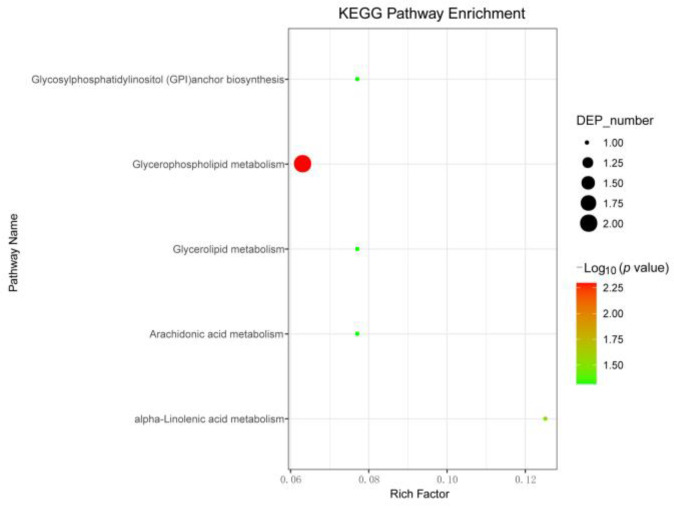
Pathway analysis associated with CSBV infection was carried out by MetaboAnalyst. The *Y*-axis, “Log10 (*p*)”, represented the transformation of the original *p*-value calculated from the enrichment analysis. The *X*-axis represents the pathway impact value calculated from the pathway topology analysis, and larger bubbles represent higher pathway impact values.

**Table 1 insects-12-00703-t001:** Primers used in reverse transcription PCR and quantitative RT-PCR.

Name	Aberration	Accession No.	Primer	Sequence (5′–3′)
phospholipase B1	AcPLB1	XM_017059237.2	F	TTGACTGACAATGGATATTATG
			R	GTCTTTCAGAAGTAGGACACAA
serine palmitoyltransferase 1	AcSPT1	XM_017049187.1	F	AAGGATTGGATGCAACTAAAGC
			R	CTGGTAATGGACAGATATTTCCAG
Ceramide synthase5	AcCS5	XM_017049042.1	F	AATAGTGCCTATGTTCCCAGCA
			R	ATCCTCACTGCTACTACTACGA
Ceramidase	AcCeramidase	XM_017052678.1	F	GCACAACGGTTACTATCGTTAC
			R	CATGTAGCTATATATTAAGGAG
Sphingomyelinase	AcSMase	XM_017050328.1	F	GTGGAGTTATTGGATCTGGACT
			R	TTGAAGTTTACAAAGTCCGACA
sphingomyelin phosphodiesterase	AcSMPD	XM_017055261.1	F	CCGTTATGCTGCAGATTGACAA
			R	GAGCATATGTTCAACGGTTCTC
sphingomyelin phosphodiesterase 1	AcSMPD1	XM_017060020.1	F	GGAATGGTGGTAGCATAACAGC
			R	TTTGCTGCTATAGAGAGCCAAT
sphingomyelin phosphodiesterase 4	AcSMPD4	XM_017064091.1	F	ATGAGGACAGATCTTGTAGCCC
			R	CGTACAATTGAAGCCCATTGAT
glucosylceramidase-like	AcGSC	XM_017065397.1	F	ATTTAAGATTCTTTAGCGCCGC
			R	GAATGGCTCGTTTCCAGTTGAA
glucosylceramidase 4	AcGSC4	XM_017065381.1	F	TCCACCGATTTTTCTACGAGAA
			R	GCTGCGAACAATAGTACTTCAG
Sacbrood virus CSBV-LN	qVp1	gi|307148859|	F	TAGAGTTACGTTTTGATTTTGTTT
			R	GCGCTAGCCGTATTTCTC

**Table 2 insects-12-00703-t002:** Lipid markers detected by Principal Component Analysis from the CSBV-infected larvae.

Significant Lipids	Theoretical Mass	Trend in CSBV vs. Control	Detection Mode	Lipid Class	Log2 (FC) aCSBV_24 h vs. CK_24 h)	Pvalue (CSBV_24 h vs. CK_25 h)	VIP (CSBV_24 h vs. CK_24 h)	Log2 (FC) a (CSBV_48 h vs. CK_48 h)	Pvalue (CSBV_48 h vs. CK_48 h)	VIP (CSBV_48 h vs. CK_48 h)
Cer (d14:1 + hO/21:0 + O)	601.551	Up	pos	Cer	1.282	0.012	2.067	1.674	0.020	1.770
Cer (d17:0/16:0 + O)	542.514	Up	pos	Cer	3.809	NS	0.875	2.358	0.0006	1.966
Cer (d20:1)	359.326	Up	pos	Cer	0.931	NS	0.250	3.093	0.043	1.656
Cer (d40:1 + hO + O)	652.588	Up	neg	Cer	0.789	NS	0.796	2.642	0.046	1.642
DG (41:0e)	703.657	Up	pos	DG	1.251	0.011	2.071	2.252	0.0001	1.989
MG (36:1)	631.563	Up	pos	MG	1.607	NS	1.698	2.281	0.004	1.904
PA (38:2)	727.528	Up	neg	PA	2.552	NS	1.138	13.545	0.035	1.694
PC (18:1/14:1)	774.529	Up	pos	PC	2.735	0.036	1.915	2.100	NS	1.138
PE (18:0/18:2)	744.553	Up	pos	PE	4.856	0.017	2.026	17.333	NS	0.987
PE (18:0e/18:3)	726.544	Up	neg	PE	1.200	0.044	1.873	1.270	0.006	1.872
PE (29:0/11:4)	802.593	Up	pos	PE	2.379	0.017	2.028	7.548	NS	1.057
SM (d20:0/19:1)	817.644	Up	pos	SM	1.572	0.024	1.980	1.455	0.006	1.873
SM (d37:1)	745.621	Up	pos	SM	1.613	0.032	1.933	1.685	0.019	1.776
SM (d41:2)	799.668	Up	pos	SM	2.641	NS	1.298	4.188	0.0317	1.708
TG (16:0/18:1/18:3)	872.770	Up	pos	TG	1.447	0.009	2.0974	1.133	0.028	1.729
TG (18:1/18:1/18:3)	898.785	Up	pos	TG	2.289	0.011	2.078	1.276	NS	1.006
TG (18:1/20:4/21:0)	951.837	Up	pos	TG	1.219	0.032	1.938	1.401	0.014	1.811
TG (24:4)	485.287	Up	pos	TG	2.307	0.029	1.953	0.651	NS	0.897
TG (39:6)	675.517	Up	pos	TG	10.899	0.049	1.848	23.588	NS	0.988
TG (43:7)	729.563	Up	pos	TG	1.251	0.031	1.943	1.383	0.015	1.804
OAHFA (16:0/26:0)	649.614	Down	neg	FA	1.361	NS	1.341	1.589	0.008	1.856
OAHFA (16:0/28:0)	677.645	Down	neg	FA	1.257	NS	1.354	1.617	0.007	1.863
OAHFA (18:1/26:0)	675.629	Down	neg	FA	1.433	NS	1.409	1.673	0.002	1.926

Log2 (FC): Log2-transformed fold change where FC = Fold Change (CSBV/Control); pos and neg represent positive mode and negative mode respectively; NS: non-significance. Lipids that putatively annotated and matched the fragmentation pattern with the database.

## Data Availability

All lipids species data are available in the Appendix A. The morphology of larvae and qRT-PCR data are available via contact to corresponding author.

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
