# Peer review of "Lipidomic Profiling Reveals Distinct Differences in Sphingolipids Metabolic Pathway between Healthy Apis cerana cerana larvae and Chinese Sacbrood Disease"

_insects, 2021, doi:10.3390/insects12080703_

Round 1
Reviewer 1 Report
In general, I am happy with the authors responses to my comments and think that the revised manuscript is now much clearer. In light of the authors responses and modifications made to the manuscript, I have no more additional comments.
Author Response
In general, I am happy with the authors responses to my comments and think that the revised manuscript is now much clearer. In light of the authors responses and modifications made to the manuscript, I have no more additional comments.
Response: We appreciate this comment and thank you.
Reviewer 2 Report
The manuscript presented by Xiaoqun Dang et al., analyzed the lipidome profile of infected A.cerana cerana larvae by CSBV in comparison with healthy ones, which is a new subject to study honey bee viruses. I recommend to accept the manuscript, but there have been a few issues through out the manuscript that I hope authors to answer them:
Material and methods: did you check your experimental samples for other diseases and viruses? If so, any results? If not, how you are sure the obtained results are due to CSBV infection?
How many colonies were used to collect brood from for the experiment?
Line 33: either delete the “at present”, or “currently”.
Line 34: the word “lipidomic” is mainly used for the method, you may need to replace it with “lipidome”.
Line 35: You may need to add the word “that” as “….2,164 lipids that were detected….”
Line 36: Lipidome?
Line 43: the word “that” repeated twice.
Line 43: Lipidome?
Line 44-45: “… in future may help to better understand the ……”
Line 54: “Infection by SBV is lethal to honey bee larvae which result entire colony for A. cerana cerana” would you please provide citation(s)?
Line 56: “thus Chinese sacbrood virus is the most serious threaten to bee health”, when you write bee health in can cover wild bees as well which is not correct. You may need to rewrite your sentence.
Line 59: “CSBV mainly infects 1- and 3-days old….” In the same sentence, what do you mean by breeding? You need to provide citation for your claims.
Line 69-70: would you please provide one more sentence explaining your claim as well as adding citation?
Line 72: you may add the “For example” before your sentence. For example, HCV life cycle…
Author Response
(1) Material and methods: did you check your experimental samples for other diseases and viruses? If so, any results? If not, how you are sure the obtained results are due to CSBV infection?
Response: We thank the reviewer for the questions. We apologize for this confusion. We did not check other diseases except CSBV. In this experiment, we have collected larvae from healthy bee colonies and transferred to the 24-well plate. One larvae per well.The larvae then was reared in the incubator at 34±1℃ and 90%± 5% relative humidity as previously described. After 12h incubation, the survived larvae was fed to the purified CSBV. The CK group was also reared in the same conditions. And then the VP1 expression level were detected in both CSBV infected larva and CK group.So we are very sure the lipidomics data obtains here are due to CSBV infection. We now have stated this clearly(line 99-line 106).
(2) How many colonies were used to collect brood from for the experiment?
Response: We thank the reviewer for the questions. Because of the lipidomics need 3 times repeat, so we have totally collected 3 colonies for this study. For each time, we have collected more than 200 larvae from one colony.
(3) Line 33: either delete the “at present”, or “currently”.
Response: Thank you. We have delete “at present” in line 33.
(4) Line 34: the word “lipidomic” is mainly used for the method, you may need to replace it with “lipidome”.
Response: Thank you. We have revised the lipidomic and lipidome through the whole manuscript.We sincerely apologize for the grammatical errors.
(5) Line 35: You may need to add the word “that” as “….2,164 lipids that were detected….”
Response: Thank you. We have added “that ” as “….2,164 lipids that were detected….” in line 35.
(6) Line 36: Lipidome?,Line 43: the word “that” repeated twice.Line 43: Lipidome?
Response: Thank you. We have delete “that” in line 43. We have change “lipidomics”in the line36 and line43 to the “lipidome”.
(7) Line 44-45: “… in future may help to better understand the ……”
Response: Thank you. We have added “help to ” in line 44.
(8) Line 54: “Infection by SBV is lethal to honey bee larvae which result entire colony for A. cerana cerana” would you please provide citation(s)?
Response: Thank you. Sorry for missing the reference. We have added the reference “Zhang,G.Z., Han,R.C. Advances on Sacbrood of Honeybees. Chinese Joumal of Biological control, 2008,24: 130-137. (in Chinese).”
(9) Line 56: “thus Chinese sacbrood virus is the most serious threaten to bee health”, when you write bee health in can cover wild bees as well which is not correct. You may need to rewrite your sentence.
Response: We thank the reviewer for the advice. We revised the sentence as “thus Chinese sacbrood virus (CSBV) is the most serious threaten to A. cerana cerana health”.
(10) Line 59: “CSBV mainly infects 1- and 3-days old….” In the same sentence, what do you mean by breeding? You need to provide citation for your claims.
Response: Thank you. Sorry for making confuse. The breeding here means that the breed bees breed larvae in the colony. We provide the reference “ Zhang,G.Z., Han,R.C. Advances on Sacbrood of Honeybees. Chinese Joumal of Biological control, 2008,24: 130-137. (in Chinese) ” to make it clear.
(11) Line 69-70: would you please provide one more sentence explaining your claim as well as adding citation?
Response: Thank you. Sorry for making confuse. We revised the sentence as that: The structural analysis of SBV showed that the pore of empty virion particle is expansion at pH 5.8 compared with the pore of full virion at pH7.4(12 Å VS 7 Å in diameter ), subsequently, the genome release[14].
(12) Line 72: you may add the “For example” before your sentence. For example, HCV life cycle…
Response: Thank you for the advice. We have added “For example” before the sentence “HCV life cycle is tightly linked to the host cell lipid metabolism”.
Reviewer 3 Report
The manuscript submitted by Dang and colleagues reported an interesting analysis of changes in lipid profiles in honey bee larvae affected with Chinese SBV.
I suggest emphasizing the result of this assay. I didn't find any relationship between the virus infection and the modification of lipidic profile changes. These allow problems in the pupation, for example? not surviving? etc. Please define better.
However, the text should be revised with attention because there are a lot of grammatical errors and typos. Also, there are some words in italics that were reported in italics, and vice-versa. Please revise it.
Finally, the English should be revised, because there are some sentences very confused and unclear.
Author Response
Thank you for your comments and suggestions.
(1) I suggest emphasizing the result of this assay. I didn't find any relationship between the virus infection and the modification of lipidic profile changes. These allow problems in the pupation, for example? not surviving? etc. Please define better.
Response: We thank the reviewer for the advice. For the pupation, the healthy larvae can finish the pupation while the CSBV infected larva can not. In this experiment, we just collected larvae at post infection 48h, and at that time, both the healthy larva and CSBV infected larva were about 4-5 days old. Besides, we have also investigated the survive rate with the same copies of CSBV infection, although not in this experiment, almost 70% larvae were dead at the post infection 96h. We provided the morphological characteristics of CSBV infected larva in Figure 1. We can definitely sure that the lipidic profile were changed under CSBV infection. The results about this section were revised as following:
To profile the alterations in the lipidomics of the A. c. cerana larvae during CSBV infection, the morphology of CSBV infected larva were displayed that the body color of CSBV infected larva was became from white to light yellow. In addition, the infected body was swollen and the ecdysial fluid accumulation under the transparent epidermis (Figure 1A and 1B). Subsequently, individual group were tested for CSBV VP1 by qRT-PCR (Figure 1C) after collecting samples to estimate virus infection per group.The relative transcript level of the VP1 was undetectable in healthy larvae. In contrast, the significant increase of the virions were found at the post-infection 12h, 24h and 48h. Both the symptoms of CSBV disease and the copies of CSBV indicating that the lipidomics profile of the larvae could represented the infected state.
Figure 1. Morphological characteristics of CSBV infected larvae and relative quantification analysis of CSBV in each time point after larvae infection.The CSBV infected larva (B) was swollen and smaller compared with the healthy larvae of A. cerana cerana (A); (C)The copies of CSBV virus.
(2) the text should be revised with attention because there are a lot of grammatical errors and typos. Also, there are some words in italics that were reported in italics, and vice-versa. Please revise it. The English should be revised, because there are some sentences very confused and unclear.
Response: We sincerely apologize for the grammatical errors and unclear presentation. We try our best to revise the manuscript including the italics and the grammatical errors. We hope this version will be acceptable.

This manuscript is a resubmission of an earlier submission. The following is a list of the peer review reports and author responses from that submission.
Round 1
Reviewer 1 Report
This article presents the first comprehensive analysis of lipid homeostasis of A. cerana cerana larva during CSBV infection through high-resolution mass spectrometry.
Comparison of lipidomics between healthy and CSBV infected larvae showed that 266 lipid species were changed by the infection of CSBV. This study provides evidence that CSBV infection specifically alters the lipid repertoire.
The lipidome of larva remodeling was significantly associated with honeybee-pathogenic CSBV infection and sphingolipids may be important for CSBV infection in A. cerana cerana larva. Specific changes in the lipidome maybe identified as possible targets for detection CSD.
In addition, qRT-PCR showed that some of genes related to ceramide metabolic pathway were up-regulated by the infection of CSBV. These data suggested that some specific lipids are required for virus infection.
Future investigations should be directed to understanding the mechanisms of how these lipid species play a role in CSBV replication, as well as their potential use as therapeutic targets for CSBV infection.
As indicated above, the authors carried out an outstanding investigation whose results are clearly presented.
I recommend its publication.
Reviewer 2 Report
Please add copies of CSBV virus, you might use the absolate abundance of RT-PCR.
Reviewer 3 Report
The manuscript “Lipidomic profiling of Apis cerana cerana larvae with Chinese sacbrood virus infection” presented a very interesting work about the lipidomic of the CSBV-larva interaction through high resolution mass spectrometry.They found 266 lipid species were changed by the infection of CSBV. Further, they provide qRT-PCR results to verify that CSBV infection specifically alters the ceramide metabolic pathway. I believe that the data set generated by the authors of this study can be useful to understanding the abnormal lipid metabolism associated with CSBV disease.
Overall, the context of this article is well organized and the significant finding is presented pretty intuitionisticly. I prefer to publish this paper after some minor revisions need to be done. My main concerns include the several points as bellowing.
(1)First, the title suggests the lipid profiles of CSBV disease. However, I believe this could only be achieved if subjects are followed over the period of time and lipid profiling is performed at various time points. Considering only two time points detected and results, it could not be show whole lipids profile. At the best, this demonstrates the differences in lipid profiles between healthy and CSBV disease.
(2) in the section about data processing,there is no information about how to identify the features. Please explain more detail about the screening criteria of 266 features .
(3)In fig.1. the title of Y-axis should be relative expression level of CSBV VP1. In fig.8, the title of X axis should be consistent with the materials, CSBV 24h should be change to CSBV_24h.
(4) “These 266 lipid molecular species distributed into ceramide(Cer), lysophosphatidylcholine(LPC), lysophosphatidylethanolamine(LPE), lysophosphatidylglycerol(LPG), monoglyceride(MG), phosphatidate(PA), phosphatidylcholine(PC), phosphatidylethanolamine(PE), diglyceride(DG), phosphatidylglycerol(PG), phosphatidylinositol(PI), phosphatidylserine(PS), sphingomyelin(SM), sphingolipids(So),triglyceride(TG) and (O-acyl)-1-hydroxy fatty acid(OAHFA) (Fig.3B)”, here fig.3B did not show information mentioned above. Further, according to fig.3B,190 out of 266 features that had higher abundances upon CSBV infection presented in text. But I can only find 182 features, please check data and describe clearly.
(5)The text describing that the raw MS data should be upload as supplement file for readers.
(6) the limitation of this work should be discussed.
(7) There are a few typos and grammar errors in this paper. The size of “ 3.2 Glycerolipids were most remodeling after CSBV infection” and 3.3,3.4 are not as the same as 3.1. The line spacing are not uniform such as section 2.5 and 3.4. It is noted that your manuscript needs careful editing by someone with expertise in technical English editing paying particular attention to English grammar,spelling and sentence structure.
(8)In addition,the list of reference is not in the journal style. It is close but not completely correct.